# Adder Attention for Vision Transformer

**Han Shu**[1]* **Jiahao Wang**[2]* **Hanting Chen**[1,3] **Lin Li**[4] **Yujiu Yang**[2] **Yunhe Wang**[1]†
[1]Huawei Noah's Ark Lab    [2]Tsinghua Shenzhen International Graduate School
[3]Peking University    [4]Huawei Technologies
{han.shu, yunhe.wang, lilin29}@huawei.com, wang-jh19@mails.tsinghua.edu.cn
htchen@pku.edu.cn, yang.yujiu@sz.tsinghua.edu.cn

## Abstract

Transformer is a new kind of calculation paradigm for deep learning which has shown strong performance on a large variety of computer vision tasks. However, compared with conventional deep models (*e.g.*, convolutional neural networks), vision transformers require more computational resources which cannot be easily deployed on mobile devices. To this end, we present to reduce the energy consumptions using adder neural network (AdderNet). We first theoretically analyze the mechanism of self-attention and the difficulty for applying adder operation into this module. Specifically, the feature diversity, *i.e.*, the rank of attention map using only additions cannot be well preserved. Thus, we develop an adder attention layer that includes an additional identity mapping. With the new operation, vision transformers constructed using additions can also provide powerful feature representations. Experimental results on several benchmarks demonstrate that the proposed approach can achieve highly competitive performance to that of the baselines while achieving an about $2$~$3\times$ reduction on the energy consumption.

## 1 Introduction

Transformers [27] have prevailed in natural language processing (NLP) due to its superior capability in capturing long-distance dependencies based on self-attention mechanism. Some homologous large-scale models, *e.g.*, BERT [10] and GPT-3 [2] provide a significant performance improvement on learning powerful language representations from unlabeled text. The great breakthrough of transformers in NLP has sparked particular interest from the vision community in applying transformer to computer vision (CV) tasks such as image recognition [11, 35, 25, 26], object detection [3, 38, 33], and image generation [4, 12, 17, 15].

The high-power consumption of transformer-based models has blocked them from being deployed on mobile devices, *e.g.*, smart phone, camera, and micro-robots. Therefore, it is necessary to study efficient transformers which can be embedded on mobile devices with affordable computation resources. Wu *et.al.* [31] present an efficient Long-Short Range Attention with two groups of heads, respectively for local context and long-distance relationship modeling. Wang *et.al.* [28] decompose the original dot-product attention into smaller attentions through linear projections. Unfortunately, previous arts mainly focus on compressing and accelerating transformers for language processing tasks, which motivates us for building efficient transformers for computer vision applications.

Various attempts have been made on compressing CNN models, including quantization [13, 16], pruning [19, 30, 23] and distillation [5, 25]. Beyond these methods, Chen *et.al.* [6] propose Adder Neural Network (AdderNet) for building efficient deep learning models, which avoids massive computational cost by replacing the convolutional operation by using $\ell_1$-distance between input

---

*Equal contribution.

†Corresponding author.

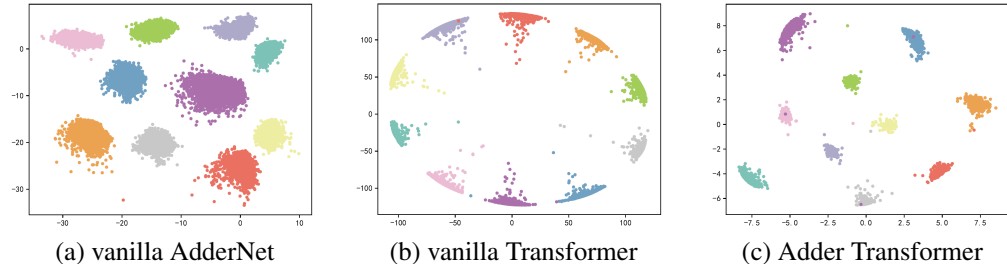

|        |        |        |
|:------:|:------:|:------:|
| (a) vanilla AdderNet | (b) vanilla Transformer | (c) Adder Transformer |

Figure 1: Visualization of features in different neural networks on MNIST dataset. From left to right are vanilla AdderNet, vanilla Transformer and Adder Transformer, respectively.

signals and weights instead of correlation. Xu *et.al.* [32] further utilize knowledge distillation technique to make AdderNet achieve better performance than CNN. Song *et.al.* [24] successfully apply AdderNet on single image super-resolution (SISR) task by learning identity mapping with self-shortcuts and implementing high-pass filter with a learnable power activation. Wang *et.al.* [29] conduct practical implementations on hardware platform (FPGA), and verify its superior performance in suppressing chip area and power consumption. Thus, we are motivated to investigate the feasibility of replacing multiplications by additions in transformer architectures.

However, it is very hard to directly transport the existing success of AdderNet on CNNs to transformers, since their basic calculation paradigm is completely different. In particular, the existing works mainly focus on the additiveization of convolutional kernel filters which do not involve the unique linear transformations and self-attention mechanisms in transformers. To this end, we propose a general adder linear transformation operation and adder self-attention mechanism to replace the traditional multiplication version in both feed-forward modules and self-attention layers in transformer model. Specifically, we utilize $\ell_1$-distance between query and key instead of scaled dot-product to measure the distance between them. For the attention module, we find that the directly using additions could cause more information concentrating on a few largest singular values, leading to a decay of the attention map rank, *i.e.*, the functionality of the attention mechanism cannot be well preserved. Thus, we propose to insert an extra identity mapping in the adder attention module to solve the decay. This operation can also homogenize the information distribution of the adder attention map. Meanwhile, we provide the detailed feed-forward and back-propagation for optimizing the adder vision transformer with stable training process. As a result, we can obtain comparable performance using the proposed adder transformer models to that of original baselines while reducing about $3\times$ of the overall energy consumption on various benchmark models and datasets.

## 2  Preliminaries and Motivation

In this section, we briefly revisit the basic related components, including AdderNet and transformer.

**Adder Neural Networks (AdderNet).** Denote filters in a convolutional layer of neural networks as $\mathbf{W} \in \mathbb{R}^{k \times k \times c_{in} \times c_{out}}$, the input feature map as $X \in \mathbb{R}^{H \times W \times c_{in}}$, and the output feature map as $Y \in \mathbb{R}^{H \times W \times c_{out}}$, where $k$ is the kernel size, $c_{in}$ and $c_{out}$ are the number of input channels and output channels, respectively. The traditional convolutional operation is defined as:

$$Y_{m,n,q} \triangleq X * F = \sum_{i=1}^{D} \sum_{j=1}^{D} \sum_{k=1}^{c_{in}} S\left(X_{m+i,n+j,k}, F_{i,j,k,q}\right). \tag{1}$$

where $S(\cdot, \cdot)$ is a pre-defined similarity measure ($S(x, y) = x \times y$ for the convolution multiplications operation). AdderNet [6] maximizes the use of additions by treating the $\ell_1$-distance between the input feature map and filter as the similarity measure (*i.e.*, $S(x, y) = |x - y|$ in Eq. 1):

$$\tilde{Y}_{m,n,q} \triangleq X \oplus F = -\sum_{i=1}^{D} \sum_{j=1}^{D} \sum_{k=1}^{c_{in}} |X_{m+i,n+j,k} - F_{i,j,k,q}|. \tag{2}$$

However the pioneering work only focused on additiveization of CNNs, while ignoring transformers.

**Vision Transformers.** Vision transformer [11] mainly consists of MHSA (Multi-head Self-Attention), FFN (Feed-Forward Network) and LN (Layer Normalization).

For multi-head self-attention, the inputs $X \in \mathbb{R}^{N \times D}$ are applied with three different linear transformations and output queries $Q \in \mathbb{R}^{M \times N \times d_q}$, keys $K \in \mathbb{R}^{M \times N \times d_k}$ and values $V \in \mathbb{R}^{M \times N \times d_v}$, where $M$ is total number of heads, $d_q$, $d_q$, $d_v$ is the dimension of queries, keys and values for each head, respectively. Generally, $D = Md_q = Md_k = Md_v = Md_h$ and the multi-head self-attention is performed as follows:

$$A_{m,j,i} = \frac{\exp\left\{\frac{Q_{m,j,:}K_{m,i,:}^T}{\sqrt{d_t}}\right\}}{Z_j}, \quad \text{where } Z_j = \sum_{i=1}^{N_{kv}} \exp\left\{\frac{Q_{m,j,:}K_{m,i,:}^T}{\sqrt{d_t}}\right\}, \quad O_{m,j,v} = \sum_{i=1}^{N_{kv}} A_{m,j,i} \cdot V_{m,i,v},$$

$$O'_{j,d} = Concat\left(O_{1,:,:}, \ O_{2,:,:}, \cdots, O_{M,:,:}\right) W_O.$$

(3)

where $d_t = d_h$, $W_O \in \mathbb{R}^{D \times D}$ denotes the weight matrix of a linear projection layer used to produce the output values $O' \in \mathbb{R}^{N \times D}$.

From the above statements, transformer has a different structure and computation paradigm compared to CNNs. Thus, it is necessary to study how to use additions to fulfill the modules.

## 3 The Proposed Model: Adder Transformer

In this section, we present the Adder Transformer— which implement the multi-head attention module and FFN module using adder operations. We will present the details in the following subsections and an illustration of the proposed method can be found in Section A.1 of the supplementary material.

### 3.1 Adder Linear Transformation

We first consider the linear transformation and FFN layers in transformers. The calculation paradigm of these two modules is based on fully connected layer, which is equivalent to a special 1x1 convolution. Therefore, we can use the original computation mechanism of adder convolutional kernel in this module. Considering weight matrices $W_Q \in \mathbb{R}^{M \times d_q \times D}$, $W_K \in \mathbb{R}^{M \times d_k \times D}$, $W_V \in \mathbb{R}^{M \times d_v \times D}$ in a projection layer of the multi-head self-attention, we maximize the use of additions by taking $\ell_1$ distance to measure the distance between the weight matrices and input embedding as:

$$Q_{m,j,q} = \sum_{d=1}^{D} -|X_{j,d} - W_{Q_{m,q,d}}|, \quad K_{m,i,k} = \sum_{d=1}^{D} -|X_{i,d} - W_{K_{m,k,d}}|, \quad V_{m,i,v} = \sum_{d=1}^{D} -|X_{i,d} - W_{V_{m,v,d}}|,$$

(4)

wherein, $m \in [1, M]$, $j \in [1, N_q]$, $i \in [1, N_{kv}]$, $q \in [1, d_q]$, $k \in [1, d_k]$, $v \in [1, d_v]$ and generally we set $N = N_q = N_{kv}$ as the resulting number of patches and $D = Md_q = Md_k = Md_v = Md_h$ as the model embedding dimension. The feed-forward network can also be modified directly:

$$FFN_a(X) = W_2 \oplus (W_1 \oplus X + b_1) + b_2,$$

(5)

As mentioned in AdderNet [6], the adder layers output values with large magnitude and should be followed by a batch normalization layer. Similarly, we apply layer normalization (LN) to scale the output values of each layer, since LN is widely used in the transformer architecture. In fact, although LN introduces multiplications, the magnitude of the multiplication can be omitted compared to the total computation of a layer. The FLOPs of adder MHSA is $DM(N_q + N_{kv})(d_k + d_v) + N_q N_{kv} M(d_k + d_v)$, and the FLOPs of adder FFN is $2N_q DrD$, where $r$ is the dimension expansion ratio. The overall FLOPs of a standard adder transformer block is $O\left(2ND(6D + N)\right)$, for regular setting $r = 4$, $N = N_q = N_{kv}$. The overall FLOPs of multiplication in Layer Normalization are $O(12ND)$. In practice, given $D = 768$ and $N = 197$, the ratio of a standard adder transformer block with the LN could be estimated as $\frac{2ND(6D+N)}{12ND} \approx 800$, which shows that LN can be easily applied in adder transformer with tiny extra energy cost.

For back-propagation, directly calculating the partial derivative of output w.r.t input and weight will lead to a sign update of $\{-1, 0, +1\}$ value, respectively. Back-propagation using the sign gradient is detrimental to the efficient updating of the parameters because the direction of the gradient can never

reach the steepest. Thus, we follow AdderNet [6] to compute the derivative of the $\ell_2$-norm with a HardTanh function to prevent gradients from exploding. Back-propagation process is formulated as:

$$\frac{\partial Q_{m,j,q}}{\partial X_{j,d}} = HT(W_{Q_{m,q,d}} - X_{j,d}), \ \frac{\partial K_{m,i,k}}{\partial X_{i,d}} = HT(W_{K_{m,k,d}} - X_{i,d}), \ \frac{\partial V_{m,i,v}}{\partial X_{i,d}} = HT(W_{V_{m,v,d}} - X_{i,d})$$

(6)

where $HT(\cdot)$ denotes the HardTanh function, and adder linear transformation in FFN modules follows the same inference and optimization process.

### 3.2 Adder Multi-Head Self-attention

#### 3.2.1 Inference process in adder multi-head self-attention

Self-attention mechanism can be viewd as the measument of the similarity of input query and key matrices. The output is usually positively correlated with the value and the correlation is determined by the magnitude of the attention score after normalization. We design the adder multi-head self-attention in strict compliance with the above principles which measure the similarity between vectors with the help of the $\ell_1$-norm, an effective measure to avoid multiplication operations. Hence, by calculating $\ell_1$-distance between each query and key vector, adder multi-head self-attention can be formulated as:

$$A_{m,j,i} = \frac{\exp\left\{\frac{-\|Q_{m,j,:} - K_{m,i,:}\|_1}{\sqrt{d_a}}\right\}}{Z_j} \ \text{ where } Z_j = \sum_{i=1}^{N_{kv}} \exp\left\{\frac{-\|Q_{m,j,:} - K_{m,i,:}\|_1}{\sqrt{d_a}}\right\}, \ d_a = 2d_t\left(1 - \frac{2}{\pi}\right),$$

(7)

wherein, $Q \in \mathbb{R}^{M \times N_q \times d_q}$, $K \in \mathbb{R}^{M \times N_{kv} \times d_k}$, $A \in \mathbb{R}^{M \times N_q \times N_{kv}}$, $m \in [1, M]$ indicates head index, $j \in [1, N_q]$, $i \in [1, N_{kv}]$ and generally we set $N = N_q = N_{kv}$ as number of patches, $d_t = d_h$ indicates embedding dimension for each head. $\frac{1}{\sqrt{d_t}}$ and $\frac{1}{\sqrt{d_a}}$ represent the scaling factor of the dot-product attention function and adder attention function, respectively.

**Theorem 1.** *Assuming that the components of $Q_{m,j,:}$ and $K_{m,i,:}$ are independent random variables following normal distribution, the variance of dot-product and $\ell_1$-distance between $Q_{m,j,:}$ and $K_{m,i,:}$ can be formulated respectively as follows:*

$$Var(Q_{m,j,:}^T K_{m,i,:}) = d_t, \ Var\left(-\|Q_{m,j,:} - K_{m,i,:}\|_1\right) = 2d_t\left(1 - \frac{2}{\pi}\right),$$

(8)

where the independent assumption is followed by [27] and the proof is provided in Section A.2 of the supplementary material. According to Theorem. 1, scaling factor $\frac{1}{\sqrt{d_t}}$ in dot-product attention is to counteract the variance explosion, and we adjust the scaling factor to match the adder circumstances.

For the process of yielding output values, the model jointly attend to information at different positions according to the attention map. Specifically, transformer model multiply the attention map with the value matrix directly to allocate each value with corresponding attention weights as in Eq. 9. For the projection to the output, we still use additions descibed in Sec. 3.1, where $W_O \in \mathbb{R}^{D \times D}$ is the weight matrix of the projection adder linear layer.

$$O_{m,j,v} = \sum_{i=1}^{N_{kv}} A_{m,j,i} \cdot V_{m,i,v},$$

$$\hat{O}_{m,j,v} = \text{LayerNorm}(O_{m,j,v}),$$

(9)

$$O'_{j,d} = -\sum_{s=1}^{D} \left| Concat\left(\hat{O}_{1,:,:}, \hat{O}_{2,:,:}, \cdots, \hat{O}_{M,:,:}\right)_{j,s} - W_{O_{s,d}} \right|.$$

#### 3.2.2 Rank Analysis on Adder Attention

In this section, we analyze the feature representation capability of adder attention by the rank of attention map. We first perform a spectral analysis of the normalized self-attention map matrix $H_t$ (Here we use $H \in \mathbb{R}^{N \times N}$ instead of $A$ to represent 2-dimensional attention matrix) and adder attention map matrix $H_a$.

Specifically, we use DeiT-Tiny model and adder DeiT-Tiny model on CIFAR-10 dataset. In order to explore the relationship between the rank of the two matrices, we apply singular value decomposition on $H_t$ and $H_a$ over different layer and heads of the two models, and plot the normalized cumulative singular value over 10000 images. As shown in Figure. 2, the cumulative singular values of both attention matrices show a clear long-tail distribution, which indicates that a few singular values dominate the matrix. Therefore, the original matrix can be replaced by a low-rank matrix with small error. In fact, the long-tail distribution effect of the adder attention matrix is more skewed than that in common self-attention matrix, indicating that the rank of the additive matrix is much lower. For example, the normalized cumulative singular value of attention matrix reached 0.9 at the 65-th largest singular value for common self-attention while at the 25-th largest singular value for adder attention, *i.e.*, main information is concentrated in the less large singular values for adder attention

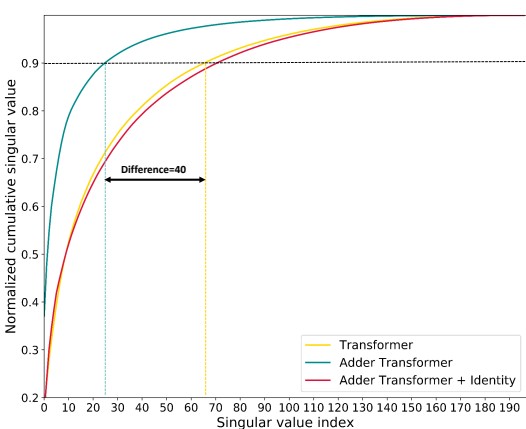

Figure 2: Spectrum analysis of the attention map in DeiT-Tiny and Adder DeiT-Tiny with $N = 197$. The Y-axis represents the cumulative normalized singular value of the attention map $H_t$ and $H_a$, and the X-axis represents the index of the eigenvalue.

and the rank of $H_a$ is lower than $H_t$. Here we make a theoretical analysis of the above results.

**Theorem 2.** *The attention matrix of adder self-attention $H_a$ can be approximated by a lower rank matrix with a certain degree of confidence than that of common self-attention $H_t$,* i.e., *when:*

$$LB\left(\Pr\left(\|\hat{H}_t v - H_t v\| \leq \epsilon_t\right)\right) = LB\left(\Pr\left(\|(\hat{H}_a \oplus w) - (H_a \oplus w)\| \leq \epsilon_a\right)\right) \quad (10)$$

*where $LB(\cdot)$ denotes the lower bound, $\oplus$ denotes the adder operation in Eq.9, $\epsilon_t, \epsilon_a > 0$, for any column vector $w, v \in \mathbb{R}^N$ of the value matrix, we have:*

$$Rank(\tilde{H}_a) < Rank(\tilde{H}_t) \quad (11)$$

*Proof.* We base on the distributional Johnson–Lindenstrauss lemma [18, 1] to finish the proof.

**Lemma 1.** *Let $R$ be a random $k \times N$ matrix, $1 \leq k \leq N$, with i.i.d. entries from $N(0, 1/k)$. For any vector $u, v \in \mathbb{R}^N$, we have:*

$$\Pr\left(\|Ru\| \leq (1 + \epsilon)\|u\|\right) > 1 - e^{-(\epsilon^2 - \epsilon^3)k/4}, \quad (12)$$

$$\Pr\left(\|u^T R^T R v - u^T v\| \leq \epsilon\|u^T v\|\right) > 1 - 2e^{-(\epsilon^2 - \epsilon^3)k/4}. \quad (13)$$

Given an approximation error $\epsilon > 0$, we define matrices $\hat{H}_t$, $\hat{H}_a$ to be the approximate low-rank matrices of $H_t$, $H_a$ respectively.

$$\hat{H}_t = H_t R^T R, \quad \hat{H}_a = S^T S H_a, \quad (14)$$

where $R$ be a random $k_1 \times N$ matrix, $1 \leq k_1 \leq N$, with i.i.d. entries from $N(0, 1/k_1)$, $S$ be a random $k_2 \times N$ matrix, $1 \leq k_1 \leq N$, with i.i.d. entries from $N(0, 1/k_2)$. According to the Sylvester's Inequality [14], we have:

$$\text{rank}(\hat{H}_t) \leq \text{rank}(H_t) = k_1, \quad \text{rank}(\hat{H}_a) \leq \text{rank}(H_a) = k_2, \quad (15)$$

In common self-attention, for row vector $x \in \mathbb{R}^N$ of matrix $\hat{H}_t$ and any column vector $v \in \mathbb{R}^N$ of the value matrix, according to the union bound [7], we have:

$$\Pr\left(\|\hat{H}_t v - H_t v\| \leq \epsilon\|H_t v\|\right) = \Pr\left(\|H_t R^T R v - H_t v\| \leq \epsilon\|H_t v^T\|\right)$$
$$\geq 1 - \sum_{x \in H_t} \Pr\left(\|x R^T R v - x v\| > \epsilon\|x v\|\right), \quad (16)$$

Then according to the second item of the Johnson–Lindenstrauss lemma, we reformulate Eq. 16 as:

$$\Pr\left(\|\hat{H}_t v - H_t v\| \leq \epsilon \|H_t v\|\right) > 1 - 2Ne^{-(\epsilon^2 - \epsilon^3)k_1/4}. \tag{17}$$

Note that Eq. 17 can be viewed as a lower bound of the reliability when using low-rank matrix in common self-attention to approximate the original attention matrix within a small error range $\epsilon$, and the lower bound is $LB_m = 1 - 2Ne^{-(\epsilon^2 - \epsilon^3)k_1/4}$ which is determined by $k_1$.

In adder self-attention, for any column vector $w \in \mathbb{R}^N$ of the value matrix, we have:

$$
\begin{aligned}
\Pr\left(\|(\hat{H}_a \oplus w) - (H_a \oplus w)\| \leq \epsilon \|H_a e\|\right) &= \Pr\left(\|(S^T S H_a - H_a)e\| \leq \epsilon \|e_a\|\right) \\
&= \Pr\left(\|e^T(H_a^T S^T S - H_a^T)\| \leq \epsilon \|e_a\|\right) \\
&> \Pr\left(\|e_a^T S^T S\| - \|e_a^T\| \leq \epsilon \|e_a\|\right).
\end{aligned}
\tag{18}
$$

where all elements in vector $e \in \mathbb{R}^N$ are fixed to 1, and we define $e_a = H_a e$. Then according to the first item of the Johnson–Lindenstrauss lemma, we reformulate Eq. 18 as:

$$\Pr\left(\|(\hat{H}_a \oplus w) - (H_a \oplus w)\| \leq \epsilon \|H_a e\|\right) > 1 - Ne^{-(\epsilon^2 - \epsilon^3)k_2/4}. \tag{19}$$

Note that Eq. 19 can also be viewed as a lower bound of the reliability when using low-rank matrix in adder self-attention to approximate the original attention matrix within a small error range $\epsilon$, and the lower bound is $LB_a = 1 - Ne^{-(\epsilon^2 - \epsilon^3)k_2/4}$ which is determined by $k_2$. When $LB_m = LB_a$ we have $k_2 = k_1 - \frac{4\log 2}{\epsilon^2 - \epsilon^3}$, which means that the approximate matrix in adder self-attention is much lower rank than that of common self-attention, then the theorem follows. $\qquad \square$

According to Theorem 2 and the above analysis, the attention matrix of the adder self-attention can be approximated by a lower rank matrix, resulting in a skewed distribution of information. To address this problem, we propose to increase the rank of the attention matrix by a more balanced distribution of singular values to attenuate the information bias in the attention map. Taking the singular value index when the cumulative normalized singular value reaches 0.9 as the equivalent rank of the attention matrix, in practice, we add an Identity matrix to each attention matrix, *i.e.*,

$$\tilde{H}_a = H_a + I. \tag{20}$$

where Identity matrix $I \in \mathbb{R}^{N \times N}$. We further investigate the effect of adding an identity mapping.

**Proposition 1.** *Denote the input attention matrix as $H \in \mathbb{R}^{N \times N}$ and an Identity mapping matrix as $I \in \mathbb{R}^{N \times N}$. The cumulative normalized singular value function $f(\cdot)$, linear function $g(\cdot)$ and their difference $D(\cdot)$ are respectively defined as:*

$$f(r) = \frac{\sum_{t=1}^r \sigma_t}{\sum_{s=1}^N \sigma_s}, \quad g(r) = \frac{r}{N}, \quad D(r) = f(r) - g(r), \tag{21}$$

*where $r = 1, 2, ..., N$, and $\sigma = [\sigma_1, \sigma_2, ..., \sigma_r, \sigma_{r+1}, ..., \sigma_N]$ ($\sigma_1 \geq \sigma_2 \geq ... \geq \sigma_N$) represents the singular value of $H$, and $D(r) \geq 0$. Denote the difference function after performing the operation $\tilde{H} = H + I$ as $\tilde{D}(r)$, and we have the following assertions:*

- *The first-order difference of $f(r)$ ($r = 1, 2, ..., N$) is non-negative and the second-order difference is non-positive, which is consistent with the concave function.*

- *$f(r)$ is closer to $g(r)$ after performing the operation: $\tilde{D}(r) \leq D(r)$.*

- *The equivalent rank of $\tilde{H}$ is higher than that of $H$: $Rank(\tilde{H}) \geq Rank(H)$.*

According to Proposition. 1 (The proof is provided in Section A.3 of the supplementary material), the distribution of singular values is more homogenized through adding identity mapping, Therefore, self-attention of each patch is reinforced to enable a more balanced distribution of information in attention map, which increases the rank of the attention matrix. As a result, more information is recovered in the attention matrix, and is correspondingly beneficial for the model to capture.

### 3.2.3  Back-propagation in adder multi-head self-attention

Back-propagation of the attention layer needs to be done with two parts of gradients, namely the partial derivative of the output w.r.t value and normalized attention score and the partial derivative of attention w.r.t query and key, respectively. Denote $A \in \mathbb{R}^{M \times N_q \times N_{kv}}$ as the attention map derived from query and key and $\hat{A} \in \mathbb{R}^{M \times N_q \times N_{kv}}$ as the attention map after softmax function. In self-attention layers, the above partial derivatives are calculated as:

$$\frac{\partial A_{m,j,i}}{\partial Q_{m,j,q}} = \frac{1}{\sqrt{d_h}} K_{m,i,q}, \quad \frac{\partial A_{m,j,i}}{\partial K_{m,i,k}} = \frac{1}{\sqrt{d_h}} Q_{m,j,k}, \quad \frac{\partial O_{m,j,v}}{\partial V_{m,i,v}} = \hat{A}_{m,j,i}, \quad \frac{\partial O_{m,j,v}}{\partial \hat{A}_{m,j,i}} = V_{m,i,v}, \quad (22)$$

For adder attention layer, we directly calculate the partial derivative of attention w.r.t query and key to conduct to a sign update. We also tried the HardTanh gradient described in Sec. 3.1, but the sign gradient showed better performance which is described in experiment part. Thus, the back-propagation process is formulated as:

$$\frac{\partial A_{m,j,i}}{\partial Q_{m,j,q}} = \frac{1}{\sqrt{d_a}} sign(K_{m,i,q} - Q_{m,j,q}), \quad \frac{\partial A_{m,j,i}}{\partial K_{m,i,k}} = \frac{1}{\sqrt{d_a}} sign(Q_{m,j,k} - K_{m,i,k}),$$
$$\frac{\partial O_{m,j,v}}{\partial V_{m,i,v}} = \hat{A}_{m,j,i}, \quad \frac{\partial O_{m,j,v}}{\partial \hat{A}_{m,j,i}} = V_{m,i,v}, \quad (23)$$

## 4  Experiments

### 4.1  Experiments on MNIST

To illustrate the effectiveness of the proposed Adder Transformer, we first train a 6 block DeiT-Tiny [25] on the MNIST dataset. The detailed network structure is shown in the supplemental material. We use AdamW optimizer [21] with. The batch size is set as 256. The original Deit-Tiny achieves a $99.46\%$ accuracy with 0.62B multiplications and 0.62B additions. By replacing the multiplications in linear transformation and self-attention layers with additions, the proposed model achieves a $99.39\%$ accuracy with 1.24B additions and minor multiplications. We visualize features in different networks as shown in Figure. 1. In CNNs and vanilla AdderNet, the features are classified according to their angles [6] or gathered into clusters, since the convolution operation can be regarded as the cosine distance and adder operation takes $\ell_1$-distance as the similarity measurement. The features of vanilla transformer are distributed in an oval shape based on angles due to the unique computational paradigm. Adder Transformer combines the advantages of both AdderNet and transformer, and the features are gathered together while at the same time distributed in an oval shape.

### 4.2  Experiments on CIFAR

We then validate our method through the representative DeiT baselines [25] on CIFAR-10 and CIFAR-100 dataset. CIFAR-10 (CIFAR-100) dataset is composed of $50k$ different $32 \times 32$ training images and $10k$ test images from 10 (100) categories. We adopt the same data augmentation strategy as that in DeiT including random crop, random clip, Rand-Augment [8], Mixup [37] and CutMix [36] to boost the performance of baseline models following. For both model we use AdamW optimizer [21] and cosine learning rate decay policy with an initial learning rate of 0.000125. We use 5 epochs for learning rate warm-up [20] with a 0.05 weight decay rate. For all experiments, the image size is set to be 224×224. We use NVIDIA Telsa-V100 GPUs and train baseline model and corresponding adder model for same epochs using PyTorch [22] library for fair comparison. Note that the models are directly trained on CIFAR dataset from scratch instead of finetuning from a pretrained model from a larger dataset like ImageNet. The experimental results are shown in Table. 1, MNN denotes the original multiplicative network, while ANN denotes replacing the multiplications in linear transformation and self-attention layers with additions by the proposed method.

For instance, the DeiT-Base model, the Adder Transformer achieve nearly the same results ($94.47\%$ in CIFAR-10 and $74.49\%$ in CIFAR-100) with MNN ($94.83\%$ in CIFAR-10 and $74.75\%$ in CIFAR-100) with little multiplications. We further calculate the energy consumptions of different models. Values in both models are 32-bit floating numbers, and the energy consumptions for a 32-bit addition and multiplication are $0.9pJ$ and $3.7pJ$, respectively [9]. Adder Transformer can obtain an about $2.5\times$ reduction on energy consumption of the Deit-B model from 80.7B$pJ$ to 32.9B$pJ$ at the cost of little performance loss. The results in DeiT-S and DeiT-T also suggest the proposed adder transformer can also achieve comparable performance to those of their baselines with massive multiplications.

Table 1: Classification results on CIFAR-10 and CIFAR-100 datasets (Training from scratch).

| Model | Method | #Mul. | #Add. | Energy (pJ) | CIFAR-10 | CIFAR-100 |
|-------|--------|-------|-------|-------------|----------|-----------|
| DeiT-B | MNN | 17.56B | 17.56B | 80.7B | 94.83% | 74.75% |
|        | ANN | 0.48B | 34.64B | 32.9B | 94.47% | 74.49% |
| DeiT-S | MNN | 4.60B | 4.60B | 21.2B | 93.22% | 73.06% |
|        | ANN | 0.24B | 8.96B | 8.9B | 92.91% | 72.74% |
| DeiT-T | MNN | 1.25B | 1.25B | 5.8B | 92.61% | 72.58% |
|        | ANN | 0.12B | 2.38B | 2.6B | 92.38% | 72.23% |

Table 2: Classification results on ImageNet datasets (Training from scratch).

| Model | Method | #Mul. | #Add. | Energy (pJ) | Top-1 Acc | Top-5 Acc |
|-------|--------|-------|-------|-------------|-----------|-----------|
| DeiT-B | MNN | 17.56B | 17.56B | 80.7B | 81.8% | 95.6% |
|        | ANN | 3.32B | 31.80B | 40.9B | 80.4% | 94.3% |
| DeiT-S | MNN | 4.60B | 4.60B | 21.2B | 79.9% | 95.0% |
|        | ANN | 0.96B | 8.24B | 11.0B | 78.3% | 93.6% |
| DeiT-T | MNN | 1.25B | 1.25B | 5.8B | 72.2% | 91.1% |
|        | ANN | 0.30B | 2.20B | 3.1B | 70.5% | 89.9% |

## 4.3  Experiments on ImageNet

We also conduct experiments on ImageNet dataset. ImageNet is a large scale vision dataset which consists of $1.2M$ different $224 \times 224$ pixel training images and $50k$ test images from 1000 different categories. We use the same data augmentation and data pre-processing method as that of DeiT. Networks are trained for 600 epochs with an initial learning rate of 0.0005 and a cosine learning rate decay. We use 5 epochs for learning rate warm-up [20] with a 0.05 weight decay rate, and the experiments are conducted on NVIDIA Tesla-V100 GPUs. The first and last block remain the multiplication. Experimental results are shown in Table. 2. Adder Transformer on DeiT-S achieves a $78.3\%$ top-1 accuracy and $93.6\%$ top-5 accuracy with 0.96B multiplications and 8.24B additions, which is slightly lower than the DeiT-S baseline in terms of accuracy but substantially reduce the energy-inefficient multiplications. The detailed energy costs are also reported in Table. 2. All models trained on ImageNet can reduce the energy cost for processing a $224 \times 224$ image by a factor of about $2\times$, which demonstrates that our models using additions in intermediate blocks can also achieve comparable accuracy to those of their baselines with massive multiplications.

## 4.4  Ablation Study

In this part, ablation studies are conducted to evaluate the effectiveness of the proposed adder linear transformation and adder self-attention. All experiments in this part is done with DeiT-Tiny network on the CIFAR dataset.

First we explore the effect of the different gradient calculation methods in back-propagation of different parts and the identity mapping, as shown in Table. 3. A total of five different settings were evaluated in this ablation study. The results shows that the optimal gradient calculation method is to use Hardtanh gradient in linear transformation layer and sign gradient in calculating the attention map. Besides, adding the identity matrix to the attention map brings more benefit to the performance, indicating the low rank property of the adder self-attention matrix and the positive effect of identity matrix on homogenizing the distribution of singular values, as shown in Figure. 2.The attention map visulization of different models is shown in Figure. 3. It can be observed that for the original adder transformer the rank of attention matrix is low. After adding the identity matrix, the rank can be well preserved and is more close to the baseline.

To evaluate the fitness of the proposed adder linear transformation and adder self-attention with the corresponding multiplicative versions, we take a reference to the mixed model which combines different types of linear transformation and self-attention. Specifically, we study four combinations: i) Linear Transformation + Self-attention (*i.e.*, original DeiT-Tiny); ii) Linear Transformation + Adder Self-attention; iii) Adder Linear Transformation + Self-attention; and iv) Adder linear Transformation

Table 3: Impact of different components of the proposed adder transformer on CIFAR-10.

| Sign update on LT | ✓ | ✓ | | | |
|---|---|---|---|---|---|
| HT update on LT | | | ✓ | ✓ | ✓ |
| Sign update on $A$ w.r.t $Q$ and $K$ | ✓ | | | ✓ | ✓ |
| HT update on $A$ w.r.t $Q$ and $K$ | | ✓ | ✓ | | |
| Identity mapping | | | | | ✓ |
| Accuracy(%) | 90.17 | 88.63 | 89.98 | 90.31 | **92.38** |

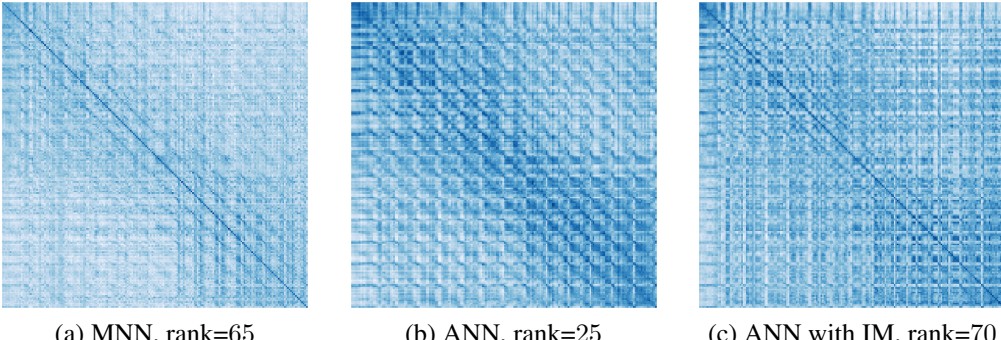

(a) MNN, rank=65          (b) ANN, rank=25          (c) ANN with IM, rank=70

Figure 3: Visualization of attention map in different networks on CIFAR-10. From left to right are Transformer, Adder Transformer w/o or with identity mapping, respectively. The equivalent rank is denoted as the singular value index when the cumulative normalized singular value reaches 0.9.

+ Adder Self-attention (*i.e.*, our Adder Transformer). For all models we train them on CIFAR-10 and CIFAR-100 to evaluate the performance and the energy cost, as shown in Table. 4. The results reveal that after either replacing adder linear transformation with the corresponding multiplication version, the performance on both datasets can have a slight improvement of $0.17\%$ and $0.25\%$ with an additional 2.9B*pJ* of energy consumption. For the case in adder self-attention, the performance on both datasets can also have a slight improvement of $0.07\%$ and $0.08\%$ with an additional 0.2B*pJ* of energy consumption. The mixed model results illustrate a good adaptability of the proposed method.

Table 4: Mixed model results on CIFAR-10 and CIFAR-100.

| Linear Transformation | Self-attention | CIFAR-10 | CIFAR-100 | Energy (pJ) |
|---|---|---|---|---|
| M | M | 92.61% | 72.58% | 5.8B |
| M | A | 92.55% | 72.48% | 5.5B |
| A | M | 92.45% | 72.31% | 2.8B |
| A | A | 92.38% | 72.23% | 2.6B |

## 5 Conclusion

This paper investigates implementing transformers using cheap addition operations. We first theoretically analyze the mechanism of self-attention and the difficulty for applying adder operation into this module. Specifically, we demonstrate the low-rank property of the attention matrix of the adder transformer models and propose to homogenize the distribution of singular values through adding an identity mapping. With the new operation, vision transformers constructed using additions can also provide powerful feature representations, which shows the potential of the addernet.

## Funding Disclosure

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
