# Adder Attention for Vision Transformer

**Han Shu**[1]* **Jiahao Wang**[2]* **Hanting Chen**[1,3] **Lin Li**[4] **Yujiu Yang**[2] **Yunhe Wang**[1]†
[1]Huawei Noah's Ark Lab    [2]Tsinghua Shenzhen International Graduate School
[3]Peking University    [4]Huawei Technologies
{han.shu, yunhe.wang, lilin29}@huawei.com, wang-jh19@mails.tsinghua.edu.cn
htchen@pku.edu.cn, yang.yujiu@sz.tsinghua.edu.cn

## A    Appendix

### A.1    Detailed Architecture

The detailed description of the Adder Transformer is given in Figure. 1. Adder Transformer takes a series of $N$ non-overlapping image patches as input. The patches are concatenated with a class token and added to positional encoding to produce a set of patch embedding. The embedding are then fed to a sequence of *Adder Multi-head Self-attention* layers and *Adder Feed-Forward Network* to model their relationship, and produces the final set of predicted class labels.

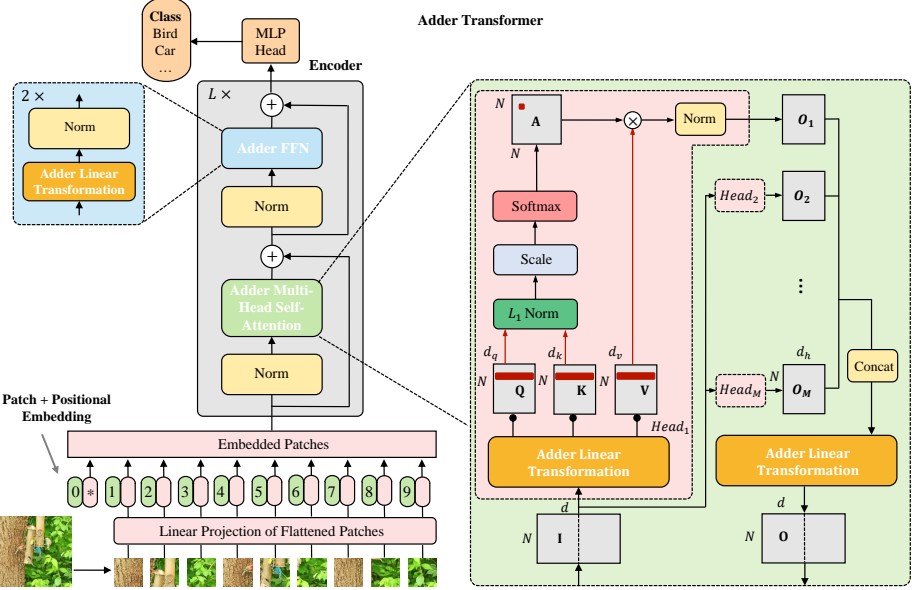

Figure 1: Illustration of the proposed Adder Transformer framework. After flattening the patch into vectors, our network alternately processes them by (1) Adder multi-head self-attention mechanism; (2) Adder feed-forward network implemented by adder linear transformation. Both with a residual connection and Pre-Layer Normalization. Red arrows indicate vector-wise operations.

---

*Equal contribution.
†Corresponding author.

## A.2 Proof of Theorem 1

*Proof.* Proof of the first equation:

Assuming that the components $\mathcal{Q} \in Q_{m,j,:}$ and $\mathcal{K} \in K_{m,i,:}$ are independent random variables following normal distribution, then both $\mathcal{Q}$ and $\mathcal{K}$ have mean 0 and variance 1, and $\mathcal{Q} + \mathcal{K}$ also follows normal distribution with mean 0 and variance 2.

Note that both $\mathcal{Q}^2$ and $\mathcal{K}^2$ follow chi-square distribution, *i.e.,* $\mathcal{Q}^2 \sim \chi^2(1)$, $\mathcal{K}^2 \sim \chi^2(1)$, then we have $\mathcal{Q}^2 + \mathcal{K}^2 \sim \chi^2(2)$. Then according to the following equation,

$$\mathcal{Q}\mathcal{K} = \frac{(\mathcal{Q} + \mathcal{K})^2 - \mathcal{Q}^2 - \mathcal{K}^2}{2}, \tag{1}$$

$\mathcal{Q}\mathcal{K}$ is actually the result of one chi-square distribution minus two chi-square distributions, and since $E\left[\chi^2(n)\right] = n$ and $Var\left[\chi^2(n)\right] = 2n$, we have $E(\mathcal{Q}\mathcal{K}) = 0$ and $Var(\mathcal{Q}\mathcal{K}) = 1$. Thus, for dot-product attention, $Q_{mj}K_{mi}^T$ has mean 0 and variance $d_t$.

Proof of the second equation:

The variance of the output of the $\ell_1$-distance between $\mathcal{Q}$ and $\mathcal{K}$ can be derived as:

$$Var(|\mathcal{Q} - \mathcal{K}|) = \left(1 - \frac{2}{\pi}\right) Var(\mathcal{Q} - \mathcal{K}) = \left(1 - \frac{2}{\pi}\right) \left[Var(\mathcal{Q}) + Var(\mathcal{K})\right], \tag{2}$$

Therefore the variance of $\ell_1$-distance between $Q_{m,j,:}$ and $K_{m,i,:}$ can be expressed as an accumulation of variances of its components:

$$Var\left(-\|Q_{m,j,:} - K_{m,i,:}\|_1\right) = 2d_t \left(1 - \frac{2}{\pi}\right), \tag{3}$$

Then Theorem 1 follows. Therefore, we adjust the scaling factor to $\frac{1}{\sqrt{d_t}}$ and $\frac{1}{\sqrt{d_a}}$ in the main paper respectively to counteract the variance explosion effect. □

## A.3 Proof of Proposition 1

*Proof.* In the main body we prove that the attention matrix of the adder self-attention has a much lower rank, resulting in a skewed distribution of information. Here we prove the effect of adding an Identity matrix to each attention matrix (Proposition 1). Figure. 2 reports the visualization results

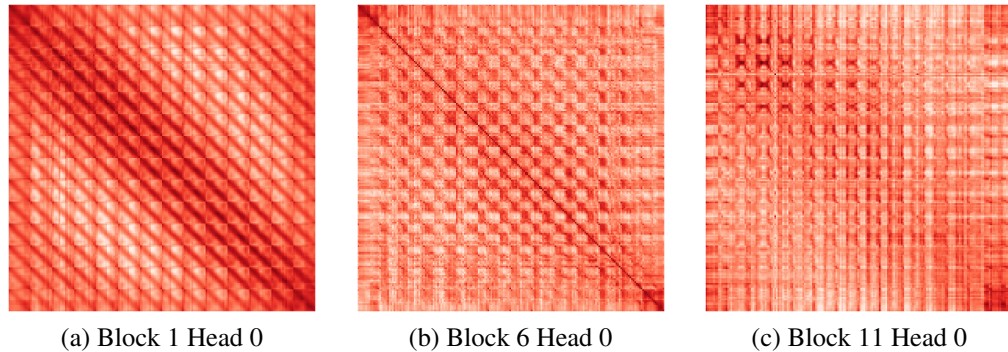

| (a) Block 1 Head 0 | (b) Block 6 Head 0 | (c) Block 11 Head 0 |

Figure 2: Visualization of attention maps of AdderTR-Tiny. From left to right are attention maps of Block 1, Head0 , Block 6, Head0, Block 11, Head0, respectively.

of the attention maps in Adder DeiT-Tiny on CIFAR-10 dataset. As shown in this figure, the adder attention maps exhibit a symmetric form in different layers. Therefore, we assume that the attention matrix is symmetric in the following analysis. We denote $\sigma_a$, $\tilde{\sigma}_a$ as the singular value of $H_a$, $\tilde{H}_a$, respectively, and $\lambda_a$, $\tilde{\lambda}_a$ as the eigenvalue of $H_a^T H_a$, $\tilde{H}_a^T \tilde{H}_a$, respectively. The relationship between

$\lambda_a$ and $\tilde{\lambda}_a$ can be expressed as:

$$\tilde{\lambda}_a = \lambda\left(\tilde{H_a}^T \tilde{H}_a\right) = \lambda\left(H_a^T H_a + H_a^T + H_a + I\right)$$
$$= \lambda\left(H_a^2 + 2H_a + I\right) \tag{4}$$
$$= \lambda_a + 2\sqrt{\lambda_a} + 1$$
$$= (\sqrt{\lambda_a} + 1)^2.$$

Note that the singular value of matrix $\tilde{H}_a$ are the square root of the corresponding eigenvalue of the matrix $\tilde{H_a}^T \tilde{H}_a$, thus we have:

$$\tilde{\sigma}_a = \sigma_a + 1, \tag{5}$$

which means that the effect of adding an Identity matrix to each attention matrix is equivalent to an increase in each singular value by 1. To discuss the impact of the above singular value change, we measure the magnitude of normalized cumulative singular value at the $r$-th ($r \leq N$) largest singular value of the attention matrix. The $N$ singular values of the matrix $H_a$ and $\tilde{H}_a$ can be written as: $\sigma_a = [\sigma_1, \sigma_2, ..., \sigma_r, \sigma_{r+1}, ..., \sigma_N]$ ($\sigma_1 \geq \sigma_2 \geq ... \geq \sigma_N$) and $\tilde{\sigma}_a = [\tilde{\sigma}_1, \tilde{\sigma}_2, ..., \tilde{\sigma}_r, \tilde{\sigma}_{r+1}, ..., \tilde{\sigma}_N]$ ($\tilde{\sigma}_1 \geq \tilde{\sigma}_2 \geq ... \geq \tilde{\sigma}_N$), and we define function $f_a(r)$ and $\tilde{f}_a(r)$ as:

$$f_a(r) = \frac{\sum_{t=1}^{r} \sigma_t}{\sum_{s=1}^{N} \sigma_s}, (r \in [1, N]), \tag{6}$$

$$\tilde{f}_a(r) = \frac{\sum_{t=1}^{r} \tilde{\sigma}_t}{\sum_{s=1}^{N} \tilde{\sigma}_s}, (r \in [1, N]), \tag{7}$$

Wherein, $f_a(r)$ and $\tilde{f}_a(r)$ denote the magnitude of normalized cumulative singular value at the $r$-th ($r \leq N$) largest singular value of the attention matrix $H_a$ and $\tilde{H}_a$, respectively. According to the *Second-order conditions* [**?**] of the concave function, suppose $f$ is differentiable (*i.e.*, its *Hessian* or second derivative $\nabla^2 f$ exists at each point in **dom** $f$, which is open). Then $f$ is concave if and only if **dom** $f$ is convex and its Hessian is negative semidefinite:

$$\nabla^2 f(x) \preceq 0. \tag{8}$$

Since $f_a(r)$ is a discrete function, we use its first-order central difference instead of the first-order derivative, which is formulated as follows:

$$\nabla f_a(r) = \frac{f_a(r+h) - f_a(r)}{h} = \frac{\sum_{t=1}^{r+h} \sigma_t - \sum_{t=1}^{r} \sigma_t}{h \sum_{s=1}^{N} \sigma_s}$$
$$= \frac{\sum_{t=r+1}^{r+h} \sigma_t}{h \sum_{s=1}^{N} \sigma_s} \geq 0 \tag{9}$$

Similarly, we use its second-order central difference instead of the second-order derivative, which is formulated as follows:

$$\nabla^2 f_a(r) = \frac{f_a(r+h) - 2f_a(r) + f_a(r-h)}{h^2}$$
$$= \frac{\sum_{t=1}^{r+h} \sigma_t - 2\sum_{t=1}^{r} \sigma_t + \sum_{t=1}^{r-h} \sigma_t}{h^2 \sum_{s=1}^{N} \sigma_s} \tag{10}$$
$$= \frac{\sum_{t_1=r+1}^{r+h} \sigma_{t_1} - \sum_{t_2=r-h+1}^{r} \sigma_{t_2}}{h^2 \sum_{s=1}^{N} \sigma_s} \leq 0.$$

Then Assertion 1 follows. The ratio of the two magnitude can be formulated as:

$$R(r) = \frac{f_a(r)}{\tilde{f}_a(r)} = \frac{\sum_{t=1}^{r} \sigma_t \sum_{s=1}^{N} (\sigma_s + 1)}{\sum_{t=1}^{r} (\sigma_t + 1) \sum_{s=1}^{N} \sigma_s} = \frac{\sum_{t=1}^{r} \sigma_t (\sum_{s=1}^{N} \sigma_s + N)}{(\sum_{t=1}^{r} \sigma_t + r) \sum_{s=1}^{N} \sigma_s}, \tag{11}$$

The difference between the numerator and denominator of Eq. 11 can be written as:

$$\sum_{t=1}^{r} \sigma_t (\sum_{s=1}^{N} \sigma_s + N) - (\sum_{t=1}^{r} \sigma_t + r) \sum_{s=1}^{N} \sigma_s = N \sum_{t=1}^{r} \sigma_t - r \sum_{t=1}^{N} \sigma_s \geq 0, \tag{12}$$

Therefore we have $R(r) \geq 1$ $(r \in [1, N])$, *i.e.*, $f_a(r) \geq \tilde{f}_a(r)$ $(r \in [1, N])$, meaning that the ascending rate of the cumulative singular value slows back down after adding an Identity matrix to each attention matrix. Then we define the linear function and corresponding difference function as:

$$g(r) = \frac{r}{N}, \quad D_a(r) = f_a(r) - g(r), \quad \tilde{D}_a(r) = \tilde{f}_a(r) - g(r), \tag{13}$$

where $D_a(\cdot)$ and $\tilde{D}_a(\cdot)$ represents the difference between the cumulative normalized singular value function and linear function, and we have:

$$D_a(r) \geq \tilde{D}_a(r). \tag{14}$$

Then, adding an identity matrix will lead to a change of the curve of Adder Transformer in Figure. 3 in the main body, more closely approximating the curve of the original transformer model. Thus, we have: $f(r)$ is closer to $g(r)$ after performing the operation: $\tilde{D}(r) \leq D(r)$, then Assertion 2 follows.

Denote $r_1$ and $r_2$ that satisfies $f_a(r_1) = 0.9$ and $\tilde{f}_a(r_2) = 0.9$, respectively. Since $f_a(r) \geq \tilde{f}_a(r)$ $(r \in [1, N])$, $\nabla f_a(r) \geq 0$ and $\nabla^2 f_a(r) \leq 0$, we have $r_1 \leq r_2$. This draws the conclusion that the distribution of information in attention map is more uniform and the equivalent rank of $\tilde{H}_a$ is higher than that of $H_a$: $Rank(\tilde{H}_a) \geq Rank(H_a)$, then Assertion 3 follows. $\square$

### A.4   Visualizations of the Attention Map

In Figure 3 we show the attention maps associated with the individual 3 layers of DeiT-Tiny model and Adder DeiT-Tiny model. For each image we present two rows: the top row correspond to the three layers of the attention maps associated with the DeiT-Tiny model. The bottom row correspond to the three layers of the Adder DeiT-Tiny model. We make two observations:

- Both adder and common self-attention can effectively focus on the key information in the picture, and the attention maps from Adder DeiT-Tiny seems to focus more on the global information.

- There exists some layers in both models that clearly focuses on the interest part of the image, on which the classification decision is performed, and some layers that focus on the context of the image, or at least the image more globally. Besides, the indexing of layers focusing on the object of interest of the adder model may not be consistent with the multiplicative model.

### References

[1] L. Faybusovich. Convex optimization-s.boyd and l. vandenberghe. *IEEE TRANSACTIONS ON AUTOMATIC CONTROL AC*, 2006.

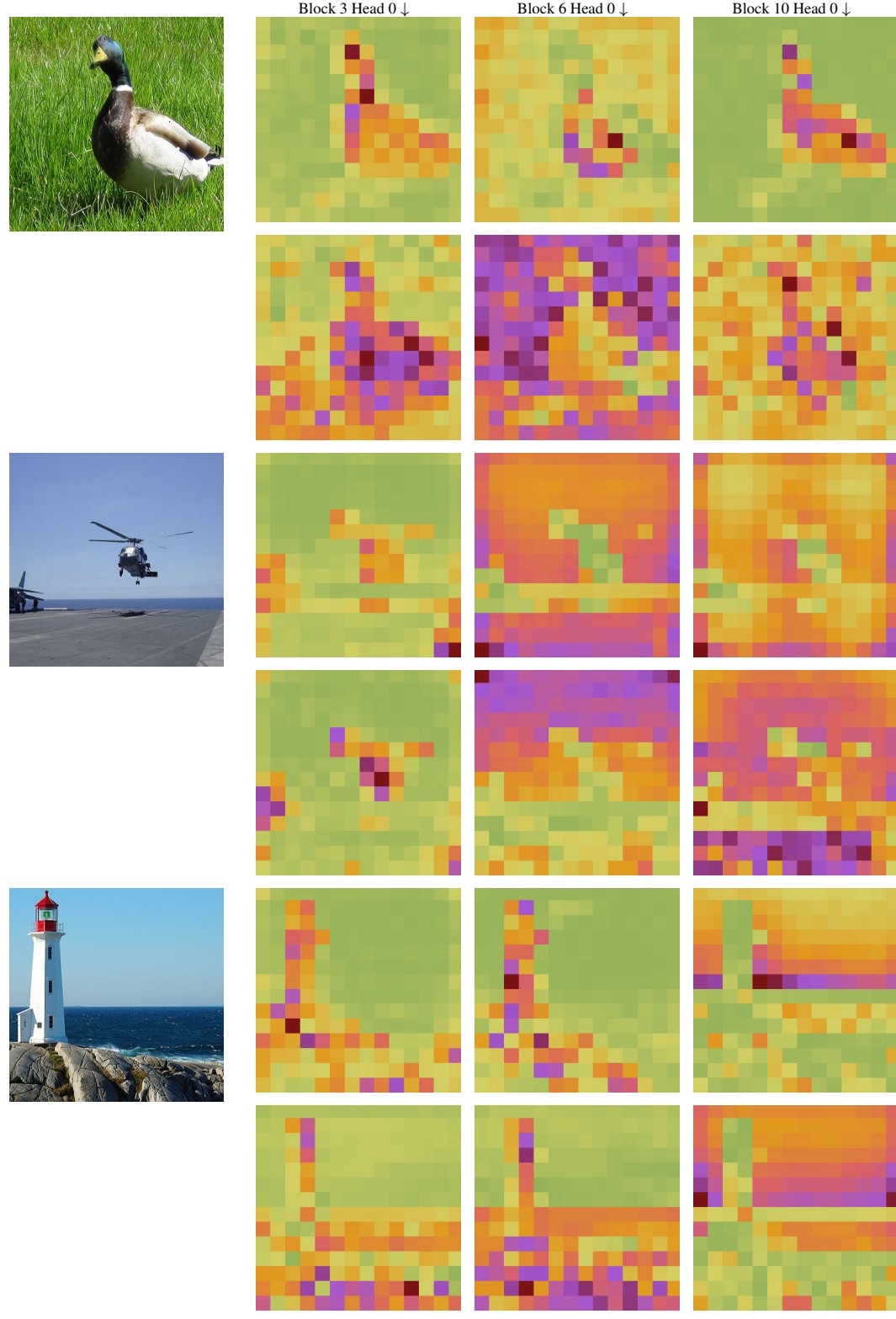

Figure 3: Visualization of the attention maps, obtained with a DeiT-Tiny model and Adder DeiT-Tiny model. For each image we present two rows: the top row correspond to the three layers of the attention maps associated with the DeiT-Tiny model. The bottom row correspond to the three layers of the Adder DeiT-Tiny model.