# OpenReview forum: "Adder Attention for Vision Transformer"
_NeurIPS.cc/2021/Conference — NeurIPS 2021 Poster_

### Official Review · Reviewer_2Zxc · 2021-07-11

**Rating:** 6
**Confidence:** 4

**Summary:**

This paper applies adder neural networks to vision transformers. Specifically, the authors propose adder attention to avoid multiplication operations and offer rank analysis and back-propagation on the adder attention.

**Limitations And Societal Impact:**

Yes

**Main Review:**

###  Originality
Although the concept of adder neural netowrk is not new, this paper applies this conect into transformers and propose adder attention to aovid multiplicaiton operations for conducting attention. The idea is okey. The author also offers analysis on adder attention.

### Quality
#### Strengths
+ The paper writes well and the proposed methods are easily to understand.
+ Beyond dot product of attention, this paper proposes new attention method under the concept of adder neural netowrk that can inspires more different methods to conduct attention.

#### Weaknesses
+ My major concern is the performance of adder attention. Taking Table 3 for example, it seems the ader neural network (ANN) is much worse than original multiplicative network (MNN), like for DeiT-S (ANN 10.5B energy 78.3% vs MNN 21.2B 79.8%). How about the performance of ANN and MNN with the same energy? DeiT-S has 384 channel dimension and 12 layers.  Maybe we can build a baseline of DeiT called DeiT-XS with 256 channel dimension and 12 layers so that DeiT-XS (MNN) has half of energy of DeiT-S (MNN), the around same as energy of DeiT-S (ANN). In this way, we can directly compare the performance of DeiT-XS (MNN) and DeiT-S (ANN).

#### Clarity
The submission is clearly written.

### Significance
Under the concept of adder nerural nentowrk, this paper propose a new attention method adder attention, but the performance of the method is not good.


**Time Spent Reviewing:**

8

---

> ### Author Response · Authors · 2021-08-09
> **Reply to Performance of ANN and MNN with the Same Energy**
>
> Sincerely thanks for your strong support and the suggested comparison will be duly addressed in our final version.
>
> **Re Performance of adder attention:**
>
> According to the comments, we conduct experiments about the performance of ANN and MNN with the same energy. DeiT-S has 384 channel dimension and 12 layers. DeiT-T has 192 channel dimension and 12 layers. We respectively build two baselines of DeiT called DeiT-XS and DeiT-XT with 256 channel dimension and 140 channel dimension respectively, both with 12 layers. As a result, DeiT-XS (MNN) and DeiT-XT (MNN) respectively has half of energy of DeiT-S (MNN) and DeiT-T (MNN). DeiT-XS and DeiT-XT are trained under the experimental setting of [1]. In this way, we can directly compare the performance of DeiT-XS (MNN) and DeiT-S (ANN), DeiT-XT (MNN) and DeiT-T (ANN). The result is shown in the bold fonts in the Table below (other results are from Table 3 in the main paper).
>
> |      Method       |   #Mul    |   #Add    | Energy(pJ) | Top-1 Accuracy |
> | :---------------: | :-------: | :-------: | :--------: | :------------: |
> |   DeiT-S (Mul)    |   4.60B   |   4.60B   |   21.2B    |     79.8%      |
> |   DeiT-S (Add)    |   0.81B   |   8.39B   |   10.5B    |     78.3%      |
> | **DeiT-XS (Mul)** | **2.14B** | **2.14B** | **9.84B**  |   **77.9%**    |
> |   DeiT-T (Mul)    |   1.25B   |   1.25B   |    6.0B    |     72.2%      |
> |   DeiT-T (Add)    |   0.22B   |   2.28B   |    3.3B    |     70.5%      |
> | **DeiT-XT (Mul)** | **0.71B** | **0.71B** | **3.27B**  |   **69.8%**    |
>
> For instance the DeiT-XS (Mul) model, it takes 2.14B multiplication and 2.14B addition operation with energy consumption of 9.84B, which is slightly lower than DeiT-S (Add) with 10.5B energy consumption, since the large scale pruning of the channel dimension. As a result, the channel dimension reduction leads to insufficient feature extraction capabilities and brings about a performance drop from 79.8% to 77.9% under the same experimental setting. As for the DeiT-XT (Mul) model, it takes 0.71B multiplication and 0.71B addition operation with energy consumption of 3.27B, which is very close to DeiT-T (Add) with 3.3B energy consumption. As a result, the large scale channel pruning leads to a performance drop from 72.2% to 69.8% under the same experimental setting.
>
> The results suggest that the superiority of Adder Transformer when ANN and MNN with comparable energy cost. Detailed results will be updated in the final version.
>
>
>
> [1] Hugo Touvron, Matthieu Cord, Matthijs Douze, Francisco Massa, Alexandre Sablayrolles, and Hervé Jégou. Training data-efficient image transformers & distillation through attention. arXiv preprint arXiv:2012.12877, 2020.

---

> > ### Comment · Reviewer_2Zxc · 2021-08-17
> > **Review to the comparision results of same energy**
> >
> > Thank the authors for supplementary results about the fair comparison regarding the same energy. To be honest, the adder attention seems to improve original multiplication attention marginally while it occupies more memory with larger channel dimension (DeiT-S (Add) with 384 channel dimension 78.3% top-1 accuracy VS. DeiT-XS (Mul) with 256 channel dimension and 77.9% top-1 accuracy). However, this paper proposes a new attention method based on the concept of adder neural networks. I still recommend leaning to accept this paper and keep my score unchanged.

---

### Official Review · Reviewer_dTzD · 2021-07-16

**Rating:** 8
**Confidence:** 4

**Summary:**

This paper presents AdderNet for Transformer. To this end, the feed-forward module and self-attention mechanism are replaced with adder operations (l1 distance). The details of forward and backward computations of AdderNet for Transformer are also provided for computer vision tasks. The proposed network is tested on CIFAR-10, CIFAR-100 and ImageNet. It's shown that the proposed network can successfully perform the image classification tasks with a small accuracy degradation while reducing the energy consumption by a factor of about 3.

**Ethical Concerns:**

I have no concern.

**Ethics Review Area:**

["I don’t know"]

**Limitations And Societal Impact:**

Yes

**Main Review:**

In general, the paper is novel and the idea of replacing multiplications with additions in Transformer is interesting specially in energy-constrained deployment scenarios.The paper is well-written and well-motivated in my opinion. Also, the accuracy results on image classification task are vigorous. My only suggestion is to make a comparison with AdderNet for CNNs in order to show trade-offs or superiority of one over another.

**Time Spent Reviewing:**

4

---

> ### Author Response · Authors · 2021-08-09
> **Reply to Comparison with AdderNet for CNNs**
>
> Sincerely thanks for your strong support and the suggested comparison will be duly addressed in our final version.
>
> **Re Comparison with AdderNet for CNNs:**
>
> We summarize the experimental results of AdderNet and Adder Transformer in the following table, which come from [1] and Table 2 in the main paper, respectively.
>
> |             Method             |  #Mul  |  #Add   | Energy(pJ) | CIFAR-10 | CIFAR-100 |
> | :----------------------------: | :----: | :-----: | :--------: | :------: | :-------: |
> |     CNN-based architecture     |        |         |            |          |           |
> |        VGG-small (Mul)         | 0.65B  |  0.65B  |   2.99B    |  93.80%  |  72.73%   |
> |        VGG-small (Add)         |   0    |  1.30B  |   1.17B    |  93.72%  |  72.64%   |
> |        ResNet-20 (Mul)         | 41.17M | 41.17M  |   0.19B    |  92.25%  |  68.14%   |
> |        ResNet-20 (Add)         |   0    | 82.34M  |   0.074B   |  91.84%  |  67.60%   |
> |        ResNet-32 (Mul)         | 69.12M | 69.12M  |   0.32B    |  93.29%  |  69.74%   |
> |        ResNet-32 (Add)         |   0    | 138.24M |   0.13B    |  93.01%  |  69.02%   |
> | Transformer-based architecture |        |         |            |          |           |
> |          DeiT-T (Mul)          | 1.25B  |  1.25B  |    6.0B    |  92.61%  |  72.58%   |
> |          DeiT-T (Add)          | 0.03B  |  2.47B  |    2.3B    |  92.38%  |  72.23%   |
> |          DeiT-S (Mul)          | 4.60B  |  4.60B  |   21.2B    |  93.22%  |  73.06%   |
> |          DeiT-S (Add)          | 0.06B  |  9.14B  |    8.4B    |  92.91%  |  72.74%   |
>
> The Table above contains mainly two parts: Experimental results for AdderNet for CNNs and Adder Transformer for Transformers.
>
> From the perspective of energy saving, AdderNet for CNNs has a slight advantage to Adder Transformer for Transformers. For AdderNet, take VGG-small for example. As a result, the AdderNets achieve nearly the same results (93.72% in CIFAR-10 and 72.64% in CIFAR-100) with CNNs (93.80% in CIFAR-10 and 72.73% in CIFAR-100) with no multiplication while achieving an about 2.5× reduction on the energy consumption. As for the ResNet-20, the convolutional neural networks achieve the highest accuracy (i.e. 92.25% in CIFAR-10 and 68.14% in CIFAR-100) but with a large number of multiplications (41.17M). The proposed AdderNets achieve a 91.84% accuracy in CIFAR-10 and a 67.60% accuracy in CIFAR-100 without multiplications, which is comparable with CNNs while achieving an about 2.6× reduction on the energy consumption. For Adder Transformer, DeiT-S (Add) can obtain an about 2× reduction on energy consumption of the DeiT-S (Mul) model from 21.2BpJ to 8.4BpJ at the cost of little performance loss.
>
> From the perspective of performance, the performance of AdderNet is better than that of Adder Transformer under the condition of the same amount of FLOPs, which is consistent with the conclusion obtained in ViT[2]. We believe that the reason is that the experimental settings are directly trained from scratch using small data sets (CIFAR-10, CIFAR-100), without involving huge data sets like JFT-300M.  This result reinforces the intuition that the inductive bias in CNN-based model is useful for smaller datasets, which shows superiority over Transformer-based model in the data-hungry condition.
>
>
>
> [1] Hanting Chen, Yunhe Wang, Chunjing Xu, Boxin Shi, Chao Xu, Qi Tian, and Chang Xu. Addernet: Do we really need multiplications in deep learning? In CVPR, 2020.
>
> [2] Alexey Dosovitskiy, Lucas Beyer, Alexander Kolesnikov, Dirk Weissenborn, Xiaohua Zhai, Thomas Unterthiner, Mostafa Dehghani, Matthias Minderer, Georg Heigold, Sylvain Gelly, et al. An image is worth 16x16 words: Transformers for image recognition at scale. In ICLR, 2020.

---

> > ### Comment · Reviewer_dTzD · 2021-08-22
> > **Review of the comparison results with AdderNet for CNNs**
> >
> > Thank you for providing comparison results with AdderNet for CNNs. Based on the results, AdderNet for CNNs is superior to Adder Transformer for classification tasks in terms of both accuracy and hardware performance. However, in the comparison between DeiT-T (Add) and VGG-small (Mul), Adder Transformer wins the race in terms of energy consumption. As a result, I still tend to accept the paper and keep my original rating.

---

### Official Review · Reviewer_KnYJ · 2021-07-16

**Rating:** 7
**Confidence:** 4

**Summary:**

This paper presents a method that converts multiplication by addition operation during the deployment of vision transformers.
**the papers main contribution**
1) The authors theoretically analyzed the mechanism of self-attention and the difficulty of applying adder operation into this module. Specifically, the feature diversity i.e., the rank of attention map using only additions cannot be well preserved.
2) Addressing previously raised difficulty, the authors develop an adder attention layer that includes an additional identity mapping, which allows vision transformers to be constructed only by additions operations.
3) The empirical results show that the proposed scheme achieves near baseline performance on several image classification datasets while achieving about 3$\times$ reduction on the total energy consumption



**Ethical Concerns:**

I didn't find any ethical concerns.

**Ethics Review Area:**

["I don’t know"]

**Limitations And Societal Impact:**

The authors adequately addressed the limitations and potential negative societal impact of their work.

**Main Review:**

This paper is mainly focused on reducing the power consumption of VIT (Vision Transformers). deployment.
For that proposed the suggest converts multiplication operations by addition operations. \
**Strengths**
The paper is a very well-written and theoretical analysis that raised the difficulties of straightforward usage of addition operations looks convincing. Also, the proposed solution insertion of the extra identity mapping in the adder attention module is well motivated. In the end, the empirical results show that the proposed scheme achieves near baseline performance on several image classification datasets while achieving about 3$\times$ reduction on the total energy consumption and specifically about 2$\times$ reduction for high energy consumption ImageNet classification VIT.

**Weaknesses**
The authors didn't provide any experiments on NLP tasks which is well known as the common usage of transformers models.

I encourage the authors to address this issue in the rebuttal phase

Post-rebuttal:

I am satisfied with the provided answers by the authors to the reviewers' concerns and I am convinced that this manuscript presents important insight to the community, hence I vote for his acceptance and keep my pre-rebuttal score.



**Time Spent Reviewing:**

3-4h

---

> ### Author Response · Authors · 2021-08-09
> **Reply to Experiments on NLP Tasks**
>
> Sincerely thanks for your constructive comments and strong support.
>
> **Re Adder attention applied on NLP tasks:**
>
> Thanks for the constructive comment.
>
> For NLP tasks, we conduct experiments on machine translation task: WMT’14 En-De, consisting of 4.5M pairs of training sentences, respectively. We apply 32K source-target BPE vocabulary, train on WMT’16, validate on newstest2013 and test on newstest2014, replicating [1]. Our baseline models are Transformer [2] with the [3] implementation. For evaluation, we use beam four and length penalty 0.6. All BLEUs are calculated with case-sensitive tokenization. We test the model with the lowest validation set loss for the task.
>
> Our training settings are in line with [1]. We apply Adam optimizer and a cosine learning rate (LR) scheduler, where the LR is linearly warmed up from $10^{-7}$ to $10^{-3}$, and then cosine annealed.
>
> In the following Table we compare various aspects of Adder Transformer with Transformer baselines.
>
> |      Method       |  #Mul  |  #Add  | Energy(pJ) | BLEU |
> | :---------------: | :----: | :----: | :--------: | :--: |
> |  Transformer[2]   | 0.338B | 0.338B |   1.55B    | 25.1 |
> | Adder Transformer | 0.112B | 0.563B |   0.92B    | 24.3 |
>
> Adder Transformer can obtain an about 1.7× reduction on energy consumption of the Transformer model from 1.55BpJ to 0.92BpJ at the cost of little performance loss on WMT’14 En-De task, compared with the baseline with massive multiplications. (Limited by time period, more Detailed results will be updated in the final version).
>
>
>
> [1] Felix Wu, Angela Fan, Alexei Baevski, Y ann Dauphin, and Michael Auli. 2019b. Pay less attention with lightweight and dynamic convolutions. In International Conference on Learning Representations.
>
> [2] Ashish V aswani, Noam Shazeer, Niki Parmar, Jakob Uszkoreit, Llion Jones, Aidan N Gomez, Łukasz Kaiser, and Illia Polosukhin. 2017. Attention is all you need. In Conference on Neural Information Processing Systems.
>
> [3] Myle Ott, Sergey Edunov, Alexei Baevski, Angela Fan, Sam Gross, Nathan Ng, David Grangier, and Michael Auli. 2019. fairseq: A fast, extensible toolkit for sequence modeling. In Proceedings of the 2019 Conference of the North American Chapter of the Association for Computational Linguistics (Demonstrations), pages 48–53, Minneapolis, Minnesota. Association for Computational Linguistics.

---

### Official Review · Reviewer_SWyG · 2021-07-16

**Rating:** 7
**Confidence:** 5

**Summary:**

 This paper designs the novel Adder paradigm for vision transformer. Specifically, the attention module is conducted with additions which saves energy consumption. This paper demonstrate the expressive power of addition computation paradigm both theoretically and experimentally on vision transformers. Experiments results show comparable accuracy on classification tasks on cifar10，cifar100 and ImageNet while saving the energy cost by 2x-3x .



**Ethics Review Area:**

["I don’t know"]

**Limitations And Societal Impact:**


 Overall I like this paper and think it made good contributions to both ViT and efficient computation fields. Some weaknesses and comments are:

1.Why the identity of the attention map can improve the expressive power of attention module? Is Identity mapping the best choice? More discussion should be conducted.
2.How about this adder attention applied on natural language processing tasks?


**Main Review:**

This is an interesting paper. The authors develop the visual transformer structures with only addition operations, where the structure and computation paradigm of attention module is carefully redesigned. This proposal can save the energy cost of transformer structures since the addition is computationally cheaper than multiplications according to previous literatures.

The author revisit the main visual transformer modules and adaptively design them in additive way. For attention module, the author discuss the possible way of realizing it with only additions and absolute operation and choose the way which both satisfy the computation constraint and the attention principle following transformers. The authors analyze the adder attention module both theoretically and experimentally. Specifically, the author conduct rank analysis on adder attention module and point out that the low rank phenomenon of the adder attention module could cause the performance decay. The decrease of attention map rank is proved theoretically and observed experimentally. To solve the issue, the author proposed to increase attention map rank by adding an identity mapping. This operation improve the map rank and the final performance.

With all the above, the ViT with adder attention achieves comparable accuracy on classification tasks on CIFAR10, CIFAR100 and ImageNet while saving 2x-3x energy cost compared with conventional structures of DeiTs.


**Time Spent Reviewing:**

11

---

> ### Author Response · Authors · 2021-08-09
> **Reply to the expressive power improvement, alternative for Identity map and NLP task experiment**
>
> Sincerely thanks for your constructive comments and strong support.
>
> **Re Why the identity of the attention map can improve the expressive power of attention module?:**
>
> Thanks for the nice concern.
>
> In section 3.2.2, we demonstrated that the rank of adder attention map is much lower than that of self-attention from the perspective of spectral analysis. Specifically, we propose and demonstrate **Theorem 1** that **the attention matrix of adder self-attention $H_a$ can be approximated by a lower rank matrix with a certain degree of confidence than that of common self-attention $H_t$.**  According to Theorem 1 and spectral analysis, the rank of adder attention map $H_a$ is lower than that of self-attention map $H_t$. Consequently, the cumulative singular values of adder attention matrices show a more skewed long-tail distribution, which indicates that a few singular values dominate the matrix. Thus, main information is concentrated in the less large singular values for adder attention compared with the traditional self-attention mechanism, resulting in a skewed distribution of information and an attention map with impeded expressive power.
>
> According to the above analysis, the key to solve the problem (*i.e.*, the low-rank problem of the adder attention map) is to increase the rank of $H_a$, that is, the distribution of the singular values of the matrix should be more balanced in order to attenuate the information bias in the attention map.
>
> In the paper, we take the singular value index when the cumulative normalized singular value reaches 0.9 as the equivalent rank of the attention matrix, and add an Identity matrix to each attention matrix. We provide **Proposition 1** (The proof is provided in Section A.3 of the supplementary material) and demonstrate that the distribution of singular values is more homogenized through adding identity mapping. Therefore, self-attention of each patch is reinforced to enable a more balanced distribution of information in attention map. As a result, more information is recovered in the attention matrix, and is correspondingly beneficial for the model to capture useful features and to improve the expressive power. After adding the Identity mapping, as shown in Figure 2 in the main paper, the cumulative normalized singular value distribution of adder attention map is closer to common self-attention map, which makes the information distribution more balanced, and the expression ability is improved and comparable to the multiplication version.
>
>
>
> **Re Is Identity mapping the best choice?:**
>
> Thanks for the constructive comment.
>
> Adding the Identity mapping is a concise and direct way to balance the information distribution of the adder attention map, but there exists other effective approaches to achieve more powerful effect.
>
> For example, following our analysis of the necessity to avoid a few singular values dominating the matrix, we propose an approach that is effective to increase the flexibility and expressive ability of the identity matrix compared to current methods for adder attention map. Formally , we add a learnable matrix on output of each adder attention map, initialized by a diagonal matrix with different setting. Adding this simple matrix after attention map is an extension of our approach and improves the training dynamic, allowing us to balance the information distribution of the adder attention map in a more fast and effective way. We refer to this approach as follows, *i.e.*,
>
> $\tilde{H_a}=H_a+Z.$
>
> where matrix $Z\in\mathbb{R}^{N\times N}$ are learnable weight parameter initialized to $\gamma \cdot I$. The matrix $Z$ is optimized along with the network during training. The diagonal values are all initialized to a small value $\gamma$, we set it to $\gamma=0.1, 0.5,  1.0$ to select the most suitable initialization.
>
> As we will show empirically, offering the degrees of freedom to do so with learnable parameter is a decisive advantage of Identity mapping over existing approaches. Adding learnable identity mapping offers more diversity in the optimization than just fixed identity matrix.
>
> In the following Table, we present an ablation study that aims at proving the effect of learnable identity mapping and finding the most suitable initialization. All experiments in this part is done with DeiT-Tiny network on the CIFAR-10 dataset, as shown in the Table.
>
> |    $\gamma$    |  0.1  |  0.5  |   1   | Fixed |
> | :------------: | :---: | :---: | :---: | :---: |
> | Accuracy($\%$) | 92.24 | 92.45 | 92.40 | 92.38 |
>
> A total of four different settings were evaluated, where the first three settings are different initialized value of the diagonal value $\gamma$ of learnable identity mapping $Z$. The final setting is directly adding an Identity mapping to the attention matrix as we did in the main paper. The results shows that the optimal performance among the four different settings is achieved by setting suitable initialization value, and directly using Identity mapping initialization can also bring further performance improvement, but it is not the best choice.
>
>
>
> **Re Adder attention applied on NLP tasks:**
>
> Thanks for the constructive comment.
>
> For NLP tasks, we conduct experiments on machine translation task: WMT’14 En-De, consisting of 4.5M pairs of training sentences, respectively. We apply 32K source-target BPE vocabulary, train on WMT’16, validate on newstest2013 and test on newstest2014, replicating [1]. Our baseline models are Transformer [2] with the [3] implementation. For evaluation, we use beam four and length penalty 0.6. All BLEUs are calculated with case-sensitive tokenization. We test the model with the lowest validation set loss for the task.
>
> Our training settings are in line with [1]. We apply Adam optimizer and a cosine learning rate (LR) scheduler, where the LR is linearly warmed up from $10^{-7}$ to $10^{-3}$, and then cosine annealed.
>
> In the following Table we compare various aspects of Adder Transformer with Transformer baselines.
>
> |      Method       |  #Mul  |  #Add  | Energy(pJ) | BLEU |
> | :---------------: | :----: | :----: | :--------: | :--: |
> |  Transformer[2]   | 0.338B | 0.338B |   1.55B    | 25.1 |
> | Adder Transformer | 0.112B | 0.563B |   0.92B    | 24.3 |
>
> Adder Transformer can obtain an about 1.7× reduction on energy consumption of the Transformer model from 1.55BpJ to 0.92BpJ at the cost of little performance loss on WMT’14 En-De task, compared with the baseline with massive multiplications. (Limited by time period, more Detailed results will be updated in the final version).
>
>
>
> [1] Felix Wu, Angela Fan, Alexei Baevski, Y ann Dauphin, and Michael Auli. 2019b. Pay less attention with lightweight and dynamic convolutions. In International Conference on Learning Representations.
>
> [2] Ashish V aswani, Noam Shazeer, Niki Parmar, Jakob Uszkoreit, Llion Jones, Aidan N Gomez, Łukasz Kaiser, and Illia Polosukhin. 2017. Attention is all you need. In Conference on Neural Information Processing Systems.
>
> [3] Myle Ott, Sergey Edunov, Alexei Baevski, Angela Fan, Sam Gross, Nathan Ng, David Grangier, and Michael Auli. 2019. fairseq: A fast, extensible toolkit for sequence modeling. In Proceedings of the 2019 Conference of the North American Chapter of the Association for Computational Linguistics (Demonstrations), pages 48–53, Minneapolis, Minnesota. Association for Computational Linguistics.

---

### Decision · Program_Chairs · 2021-09-27

**Decision:**

Accept (Poster)

**Comment:**

The reviewers are all in agreement that this paper provides novel and interesting results exploring a theoretical analysis of self-attention, and the introduction of an adder attention layer which helps reduce energy requirements. I would encourage the authors to incorporate some of the additional results on NLP tasks and CNN comparisons into their main text.